

# Safety of outpatient *vs.* inpatient anterior cervical discectomy and fusion: a systematic review and meta-analysis

Lili Ding[1], Mengzhu Yin[1] and Wenhua Yuan[2]

[1] Department of Acupuncture and Moxibustion, Guali Town Community Health Service Center (Guali Branch, General Hospital of Medical Community, Xiaoshan District, Hangzhou), Hangzhou, Zhejiang, China
[2] Institute of Orthopedics and Traumatology, the First Affiliated Hospital of Zhejiang University of Traditional Chinese Medicine, Hangzhou, Zhejiang, China

## ABSTRACT

**Objective**. To evaluate and compare the safety of outpatient and inpatient anterior cervical discectomy and fusion (ACDF) regarding complications and related outcomes.
**Methods**. PubMed, Embase, and Scopus were systematically searched for retrospective cohort studies published between January 1, 2000, and December 31, 2024, comparing outpatient and inpatient ACDF. Pooled relative risks (RRs) and weighted mean differences (WMDs) were calculated using a random-effects model. Study quality and certainty of evidence were assessed using the Newcastle–Ottawa Scale (NOS) and GRADE, respectively.
**Results**. A total of 21 studies involving 164,541 patients (36,361 outpatient and 128,180 inpatient) were included. Outpatient ACDF resulted in significantly lower incidence of overall complications (RR 0.45, 95% CI [0.35–0.57]), mortality (RR 0.35, 95% CI [0.16–0.77]), deep vein thrombosis (RR 0.56, 95% CI [0.37–0.85]), and wound complications (RR 0.59, 95% CI [0.52–0.68]). Reduced risks were also observed for unplanned reoperations (RR 0.33, 95% CI [0.24–0.46]), readmissions (RR 0.57, 95% CI [0.46–0.70]), and pulmonary complications (RR 0.43, 95% CI [0.27–0.68]). Risks of stroke, dysphagia, hematoma, and renal and cardiac complications were comparable between the groups. The certainty of evidence was rated low to very low due to high heterogeneity, retrospective study designs, and indirectness.
**Conclusion**. Outpatient ACDF is associated with fewer complications as compared to inpatient procedures for carefully selected patients. However, the retrospective nature of the studies, the possibility of selection bias, and low-certainty evidence underscore the need for high-quality prospective research to validate these results and inform clinical practice.

Corresponding author
Lili Ding, 13858036527@163.com

## INTRODUCTION

With the rising cost of health care around the world, interest has developed in reducing expenditure by several means like, use of generic drugs, remote care pathways, minimizing hospital stay, and improving healthcare financing systems (*Cohn, 2014*). Of these, remote care pathways and outpatient surgery are rapidly being incorporated in clinical

practice. Remote care *via* telemedicine encompasses consultations by videoconferencing or telephone is now being utilized for orthopedic referrals. This encompasses patients who do not necessitate a thorough physical, internal, or visual examination. Remote consultations reduce the need for medical personnel to travel in certain circumstances, thereby reducing costs (*Moldovan & Moldovan, 2025*). On the other hand, outpatient surgery provides numerous benefits, such as expedited recovery periods, increased convenience, and reduced costs. Patients frequently experience reduced tension and are able to recuperate in the privacy and comfort of their own homes. Furthermore, the more predictable schedules of ambulatory settings can be advantageous for both patients and medical personnel (*Zhang et al., 2024*).

Anterior cervical discectomy and fusion (ACDF) is routinely used to treat degenerative cervical spine conditions, including cervical radiculopathy and myelopathy (*Song & Choi, 2014*). Historically, ACDF has been performed in an inpatient setting due to concerns over postoperative complications, including dysphagia, hematoma, and respiratory distress (*Song & Choi, 2014*; *Gould, Sohail & Haines, 2019*; *Narain et al., 2020*). However, due to advancements in surgical techniques, anesthesia, and perioperative care, there is a growing interest in outpatient ACDF that may provide several advantages, such as reduced healthcare costs, lower risk of nosocomial infections, and enhanced patient satisfaction (*Erickson et al., 2007*; *Rossi et al., 2020*; *Safaee et al., 2021*). However, while outpatient ACDF is associated with potential cost savings and is generally considered safe, there is still limited data on its safety and the impact on patient outcomes.

Previous reviews and meta-analyses have highlighted the need for further synthesis and evaluation of the evidence (*Ban et al., 2016*; *McClelland et al., 2016*; *Yerneni et al., 2020*; *Epstein, 2021*). *Ban et al. (2016)* evaluated the safety of outpatient ACDF compared to inpatient surgery using a meta-analysis of 12 studies. They concluded that outpatient ACDF is a cost-effective method that is equally safe and has a similar risk of complications as inpatient surgery if postoperative complications are closely monitored (*Ban et al., 2016*). More recently, the meta-analysis of 15 studies by *Yerneni et al. (2020)* indicated no significant difference in the incidence of overall complications, stroke, dysphagia, thrombolytic events, and hematoma between outpatient and inpatient ACDF. However, rates of reoperation, mortality, as well as length of hospitalization were significantly lower in cases of outpatient ACDF. The study concluded that while outpatient ACDF can be safe for well-selected patients, those with advanced age and comorbidities may not be suitable (*Yerneni et al., 2020*). The most recent review focused on studies published up to April 1, 2018 (*Yerneni et al., 2020*). Since then, several high-quality studies with large sample sizes have been published, providing valuable evidence on a broader range of previously unexplored outcomes. This study aims to provide an updated comprehensive summary and analysis of the evidence comparing the safety of outpatient and inpatient ACDF. The research question was: Does outpatient ACDF lead to similar risk of mortality and complications as compared to inpatient ACDF?

## METHODS

This systematic review strictly adhered to the Preferred Reporting Items for Systematic Reviews and Meta-Analyses (PRISMA) guidelines (*Page et al., 2021*). The protocol was preregistered in the PROSPERO database (CRD42025636625).

### Databases searched

PubMed, Embase, and Scopus databases were searched by two reviewers (MY & WY) for relevant studies with the publication dates between January 1, 2000, and December 31, 2024, that evaluate the safety of outpatient *versus* inpatient ACDF. Bibliographies of included studies were manually searched for potentially missed studies. The search strategy used for each of the three databases is presented in Table S1.

### Inclusion and exclusion criteria

Inclusion criteria: (1) observational studies (retrospective or prospective), including registry-based analyses and randomized controlled trials; (2) studies examining populations undergoing ACDF and explicitly comparing safety outcomes of outpatient and inpatient surgical settings; (3) studies that reported key clinical outcomes such as complications, readmissions, reoperations, or other relevant measures of safety, preferably within 12 months post-operative period, although we did not impose restrictions on inclusion based on timing of assessment of complications; (4) publications in English or with a readily available English translation.

Exclusion criteria: (1) Studies published before the year 2000 or involving subjects who underwent surgery before the year 2000; (2) articles that did not feature a direct comparison of outpatient *versus* inpatient ACDF, lacked explicit outcome data, or focused solely on other types of spinal procedures; (3) conference abstracts without full-text publications, single-patient case reports, letters to the editor, and editorials.

When multiple publications presented overlapping datasets, only the most recent or most comprehensive publication was included to avoid duplication.

### Study screening and final selection process

All identified studies were deduplicated. Two reviewers (MY & WY) independently assessed each title and abstract using the predetermined eligibility criteria. The literature screening was done from 5th February until 10th February 2025. An inter-rater reliability assessment was performed during the initial screening to evaluate consistency in study selection, with a predefined acceptable threshold of 80%. If the agreement fell below this threshold, the senior author (LD) would have convened a detailed discussion to ensure clarity and alignment regarding the inclusion and exclusion criteria. However, this step was not required, as the inter-rater reliability reached approximately 92%.

For studies that passed the title and abstract screening, a full-text review was conducted to confirm final eligibility. If discrepancies arose during either phase, they were resolved through a structured consensus process. Initially, the two reviewers (MY & WY) engaged in discussion to reassess the study, carefully cross-referencing it with the eligibility criteria and prior decisions. If the discrepancy was due to unclear reporting, additional details from

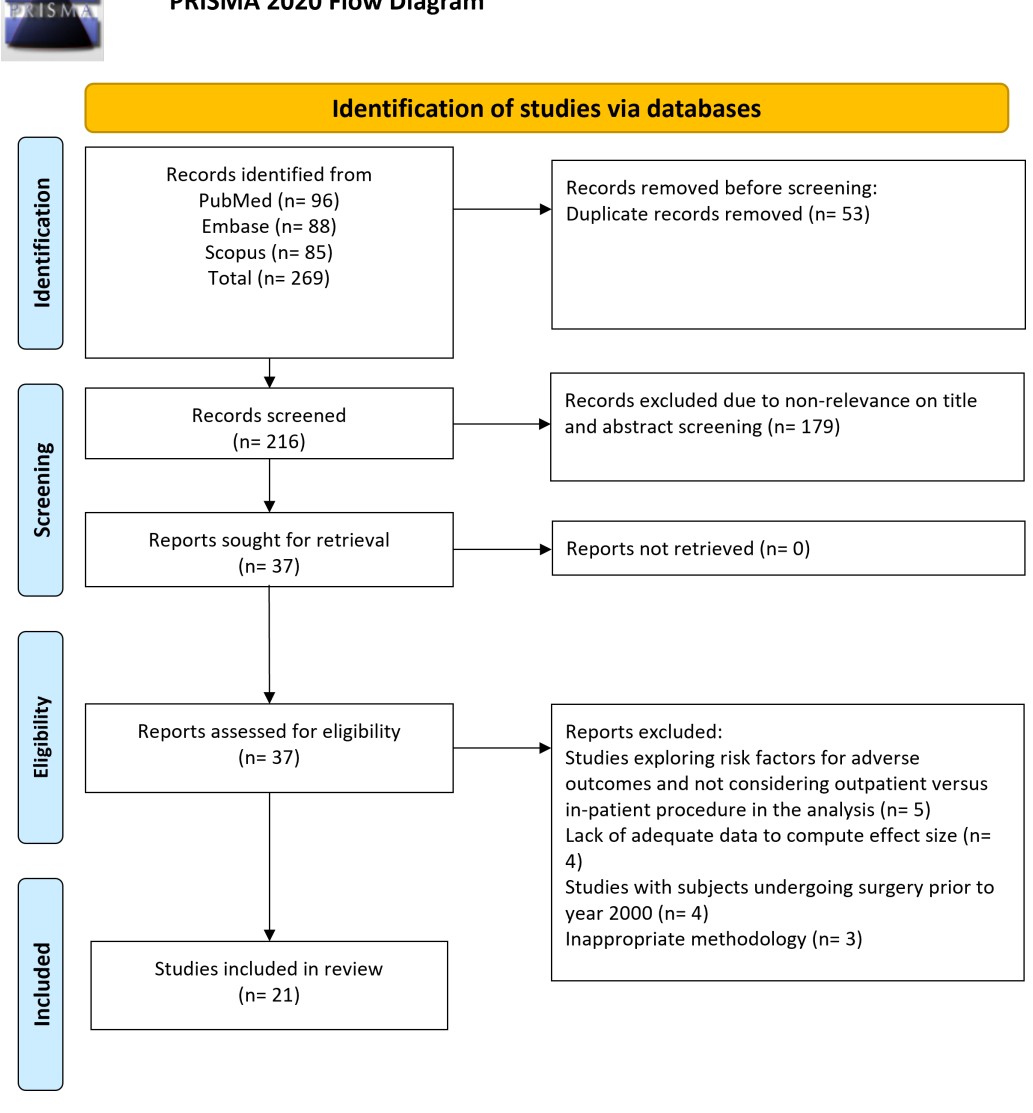

**Figure 1** Selection process of studies included in the review.

Supplementary Materials, study protocols, or cited references were reviewed to ensure an informed decision. In cases where consensus could not be reached after discussion, the senior author (LD) was consulted to provide an independent evaluation and make the final determination. PRISMA flow diagram of the study is shown in Fig. 1.

## Data extraction

Details on study characteristics (author, year, design, setting), participant demographics (age, sex, body mass index, comorbidity indices), surgical procedure (levels fused, inpatient *vs.* outpatient status), and postoperative outcomes of interest were collected using a

standardized form. Any discrepancies or missing data were resolved through discussion. The data extraction was done from 12th February till 25th February 2025.

### Quality assessment of individual studies

All observational studies were appraised using the Newcastle–Ottawa Scale (NOS) (*Wells et al., 2023*). The comparability, participant selection, and outcome assessment of each study were assessed using the questions available in NOS. The general quality of a study was demonstrated by its NOS score, which could range from 0 to 9. Higher scores meant lower chance of bias indicating more rigorous methods were used. Two reviewers (MY & WY) assessed each domain independently, and discrepancies in scoring were resolved by consensus, or by discussion with the senior author (LD).

### GRADE assessment

Certainty of the evidence for each outcome was analyzed using the GRADE approach (*McMaster University and Evidence Prime, 2024*). GRADE assesses the risk of bias, indirectness, inconsistency, imprecision, reporting bias, *etc.* The outcomes were rated as high, moderate, low, or very low certainty of evidence.

### Statistical analysis

Pooled relative risks (RRs) and weighted mean differences (WMDs) were calculated for categorical and continuous outcomes, respectively. The pooled effect sizes were reported along with 95% confidence intervals. The random effects model was used for all analyses to account for variability in patient characteristics (*e.g.*, age, sex distribution, baseline comorbidities), follow-up periods, and the methods used for assessing and reporting outcomes (*Higgins et al., 2019*). Publication bias was assessed by funnel plot symmetry and Egger's test. $P < 0.05$ indicated potential bias (*Egger et al., 1997*). All analyses were done by STATA software version 16.0. $P < 0.05$ indicated significance.

## RESULTS

Of 269 identified studies (Fig. 1), 53 duplicates were removed, and the titles and abstracts of 216 unique records were inspected. Of them, 179 studies were eliminated as not meeting the review's objectives. The full texts of the 37 studies were evaluated. Eventually, 21 studies (*Stieber et al., 2005*; *Liu, Briner & Friedman, 2009*; *Martin et al., 2014*; *McGirt et al., 2015*; *Adamson et al., 2016*; *McClelland et al., 2017*; *Fu et al., 2017*; *Khanna et al., 2018*; *Purger et al., 2018*; *Purger et al., 2019*; *Mullins et al., 2018*; *Arshi et al., 2018*; *Vaishnav et al., 2019a*; *Vaishnav et al., 2019b*; *Patel et al., 2019*; *Shenoy et al., 2019*; *Khalid et al., 2019*; *Boddapati et al., 2021*; *Lee et al., 2021*; *Kamalapathy et al., 2021*; *Tani et al., 2023*) were incorporated in the final analysis (Table 1).

The included studies provided data from 164,541 patients, of whom 36,361 were treated as outpatients, and 128,180 as inpatients. All the studies had a retrospective cohort design and were done in the United States of America (Table 1). The inpatient groups tended to be older and exhibit a higher comorbidity burden, as reflected by higher American Society of Anesthesiologists (ASA) classification (often ASA ≥3) and/or Charlson Comorbidity
**Table 1  Included studies with their key characteristics.**

| Study identifier | Design and location | Mean age, sex and BMI distribution in both groups | Comorbidity distribution and levels operated | Sample size | Assessment time post-surgery | NOS score |
|---|---|---|---|---|---|---|
| *Tani et al. (2023)* (16) | RC; USA | Patients older in inpatient group (mean age 54 *vs.* 51 years); less obese/over-weight in inpatient group (58 *vs.* 70%); lower proportion of males in inpatient group (46 *vs.* 64%) | Higher ASA class III or more in inpatient group (16 *vs.* 5%); higher CCI in inpatient group; higher proportion with two-level fusion in inpatient group (51 *vs.* 34%) | 662 (494 outpatient group; 168 in-patient group) | During hospital stay | 7 |
| *Boddapati et al. (2021)* (17) | RC; USA | Those in the outpatient group were younger (>70 years; 7% *vs.* 14%); similar sex distribution in both groups (∼50% male); similar BMI distribution | Those in the outpatient group had lower rates of diabetes (16% *vs.* 20%), dependent functional status (0.6% *vs.* 2.6%), and lower ASA classification; also, a higher proportion had a three-level fusion (84.5 *vs.* 77.5%) | 3,441 (723 outpatient group; 2,718 in-patient group) | 30 days | 8 |
| *Kamalapathy et al. (2021)* (18) | RC; USA | Propensity score matching done, so similar age (majority above 40 years) and sex (56% females) distribution among the groups; BMI not reported | Similar CCI score and comorbidity distribution among the groups; all underwent multi-level fusion | 31,154 (15,577 outpatient group; 15,577 in-patient group) | 90 days | 7 |
| *Lee et al. (2021)* (19) | RC; USA | Patients in both groups had similar age (mean 57); similar proportion of males (53%) and similar BMI (30.1 kg/m2) in both groups | Similar proportion in ASA class (ASA <3 i.e., 54%) and similar proportion with level one and two fusion (95%) in both groups | 10,384 (2,610 outpatient group; 7,774 in-patient group) | 30 days | 8 |
| *Vaishnav et al. (2019a)* (20) | RC; USA | Younger subjects in outpatient group (mean age 52.2 *vs.* 56.7 years); similar sex distribution (∼60% male); BMI was not significantly different between the 2 groups (mean around 28 kg/m2) | Higher CCI in inpatient group (mean 2.26 *vs.* 1.56); higher proportion in ASA class III in inpatient group; in-patient group had a lower proportion with two-level fusion (71.7 *vs.* 87.7%) | 103 (57 outpatient group; 46 in-patient group) | 6 months | 8 |

Ding et al. (2025), *PeerJ*, DOI 10.7717/peerj.20045

**Table 1** (*continued*)

| Study identifier | Design and location | Mean age, sex and BMI distribution in both groups | Comorbidity distribution and levels operated | Sample size | Assessment time post-surgery | NOS score |
|---|---|---|---|---|---|---|
| *Shenoy et al. (2019)* (21) | RC; USA | Those in the outpatient group were younger (mean age 46 *vs.* 51 years) and had lower proportion of males (60 *vs.* 93%); data on BMI not provided | Those in the outpatient group had higher smoking rates (25% *vs.* 14%); the mean number of levels operated was higher in in-patient group (1.6 *vs.* 1.3); no other information of comorbidities reported | 434 (126 outpatient group; 308 in-patient group) | 30 days | 6 |
| *Patel et al. (2019)* (22) | RC; USA | Patients in the inpatient group were older (53 *vs.* 48 years); lower proportion of males (57 *vs.* 65%) and higher proportion of obese (46 *vs.* 38%) in inpatient group | Patients in the inpatient group were more likely to be diabetic (16% *vs.* 5.0%) and have a higher-comorbidity burden; majority with level one or two fusion in both groups (>90%) | 272 (100 outpatient group; 172 in-patient group) | Within six months | 7 |
| *Vaishnav et al. (2019b)* (23) | RC; USA | Patients in both groups had similar age (mean 52 *vs.* 53 years); lower proportion of males (52 *vs.* 64%) and lower mean BMI (27.3 *vs.* 30.4 kg/m2) in outpatient group | Lower proportion in ASA class III in outpatient group; all had two-level fusion | 83 (25 outpatient group; 58 in-patient group) | 6 months | 7 |
| *Khalid et al. (2019)* (24) | RC; USA | Patients aged 65 years and above; proportionately more older subjects in inpatient group (≥80 years; 7 *vs.* 0%); lower proportion of males in inpatient group (47 *vs.* 51%); BMI not reported | Similar comorbidity distribution except for lower proportion with previous MI in inpatient group (4.8 *vs.* 31.3%); All with >2-level fusion | 2,492 (144 outpatient group; 2,348 in-patient group) | 30 days | 8 |

Peer/

**Table 1** (*continued*)

| Study identifier | Design and location | Mean age, sex and BMI distribution in both groups | Comorbidity distribution and levels operated | Sample size | Assessment time post-surgery | NOS score |
|---|---|---|---|---|---|---|
| *Purger et al. (2019)* (25) | RC; USA | Patients older in inpatient group (mean 46 *vs.* 42 years); lower proportion of males in inpatient group (47 *vs.* 51%); no data reported on BMI | Higher CCI in inpatient group (mean 0.25 *vs.* 0.12); levels of fusion not reported | 2,159 (370 outpatient group; 1,789 in-patient group) | 30 days | 7 |
| *Khanna et al. (2018)* (26) | RC; USA | Patients older in inpatient group (mean 53 *vs.* 50 years); similar proportion of males in both groups (50%); similar BMI (∼30 kg/m2) | Inpatient groups had a higher prevalence of chronic obstructive pulmonary disease; higher ASA scores of 3 or more (37% *vs.* 31%; *P* < 0.001), higher proportion with bleeding disorder; All with single level fusion | 6,940 (1,778 outpatient group; 5,162 in-patient group) | 30 days | 8 |
| *Mullins et al. (2018)* (27) | RC; USA | Inpatients were older (median age; 53 *vs.* 47 years), were more commonly male; similar BMI (∼29 kg/m2) | Inpatient groups had a higher rate of diabetes (19 *vs.* 11%); majority with level one or two fusion (70%) | 1,123 (560 outpatient group; 563 in-patient group) | 25 months | 6 |
| *Arshi et al. (2018)* (28) | RC; USA | Patients older in inpatient group; median age of the cohort 65 to 69 years; similar proportion of males (49%) | Higher CCI in inpatient group (mean 2.81 *vs.* 1.74); in both groups, one or two-level fusion | 12,179 (1,215 outpatient group; 10,964 in-patient group) | 12 months | 8 |
| *Purger et al. (2018)* (29) | RC; USA | Patients older in inpatient group (mean 53 *vs.* 48 years); similar proportion of males in both groups (48%); no data reported on BMI | Higher CCI in inpatient group (mean 0.37 *vs.* 0.17); in both groups, one or two-level fusion | 50,131 (3,135 outpatient group; 46,996 in-patient group) | 30 days | 7 |
| *Fu et al. (2017)* (30) | RC; USA | Patients older in inpatient group (≥65 years; 21 *vs.* 8%); similar BMI in both groups (55% non-obese in both groups); similar proportion of males (49%) | Higher CCI in inpatient group (CCI of 4 or more; 16 *vs.* 6%); in both groups, one or two-level fusion (>90%) | 21,025 (4,597 outpatient group; 16,428 in-patient group) | 30 days | 8 |

**Table 1** (*continued*)

| Study identifier | Design and location | Mean age, sex and BMI distribution in both groups | Comorbidity distribution and levels operated | Sample size | Assessment time post-surgery | NOS score |
|---|---|---|---|---|---|---|
| *McClelland et al. (2017)* (31) | RC; USA | Patients older in inpatient group (mean 51 *vs.* 48 years); lower proportion of males in inpatient group (48 *vs.* 53%) | Subjects with one or two-level fusion; no information on comorbidities | 10,080 (2,016 outpatient group; 8,064 in-patient group) | 30 days | 6 |
| *Adamson et al. (2016)* (32) | RC; USA | Those in the outpatient group were older (mean age of around 49 years *vs.* 46 years); similar proportion of males in both groups (48%) and similar BMI (mean 29 kg/m2) | Similar comorbidity distribution except for high depression and osteoporosis in the inpatient group; level-one fusion in 60% and level-two in 40% | 1,478 (994 outpatient group; 484 in-patient group) | 90 days | 7 |
| *McGirt et al. (2015)* (33) | RC; USA | Patients older in inpatient group (mean age 54 *vs.* 49 years); similar BMI in both groups (∼30 kg/m2); higher proportion of males in inpatient group (49 *vs.* 46%) | Higher comorbidities in outpatient group (diabetes, smoking status, functionally dependent, chronic obstructive pulmonary disease [COPD], hypertension and grade); 1–2-level fusion | 7,288 (1,168 outpatient group; 6,120 in-patient group) | Within 30 days | 8 |
| *Martin et al. (2014)* (34) | RC; USA | Patients older in inpatient group (mean age 52 *vs.* 49 years); similar BMI in both groups (∼29 kg/m2); higher proportion of males in inpatient group (50 *vs.* 47%) | Similar comorbidity distribution; higher proportion in ASA class III or more in inpatient group (35 *vs.* 27%); All had single-level fusion | 2,914 (597 outpatient group; 2,317 in-patient group) | Within 30 days | 8 |
| *Liu, Briner & Friedman (2009)* (35) | RC; USA | Patients older in inpatient group (mean age 56 *vs.* 49 years); high obesity in inpatient group (8 *vs.* 4%); lower proportion of males in inpatient group (58 *vs.* 69%) | All with single-level fusion; high proportion with comorbidities in the inpatient group (hypertension, high cholesterol, diabetes) | 109 (45 outpatient group; 64 in-patient group) | 63 days (mean) | 6 |
| *Stieber et al. (2005)* (36) | RC; USA | The groups were comparable in age (mean around 44 years) and body mass index (mean around 27 kg/m2); higher proportion of male in outpatient group (60 *vs.* 40%) | Similar proportion with two-level fusion in both groups (around 56%); comorbidity data not provided | 90 (30 outpatient group; 60 in-patient group) | Within 3 weeks | 6 |

**Notes.**

RC, retrospective cohort; CCI, Charlson Comorbidity Index; ASA, American Society of Anesthesiologists; MI, myocardial infarction.

Index (CCI) scores. The outpatient groups, in contrast, were generally younger, with fewer comorbidities. Many studies examined one- or two-level ACDF in both groups (17 out of 21 included studies). Where reported, BMI was comparable between groups, though a few studies indicated higher obesity rates in the inpatient group (Table 1). The postoperative outcomes were primarily assessed within 6 months post-operatively (19 out of 21 studies). Most studies demonstrated moderate-to-high methodological quality (score of 6 and above) based on their NOS assessments (Table 1; Table S2). The mean NOS score was 7.2.

## Risk of complications

Compared to the inpatient ACDF group, patients undergoing outpatient/ambulatory procedures had reduced risk of "any" complications (RR 0.45, 95% CI [0.35–0.57]; $n = 14$, $I^2 = 74.7\%$) and mortality (RR 0.35, 95% CI [0.16–0.77]; $n = 7$, $I^2 = 0.0\%$) (Fig. 2). Furthermore, these patients had a lower risk of deep vein thrombosis (DVT) (RR 0.56, 95% CI [0.37–0.85]; $n = 8$, $I^2 = 39.6\%$). However, the risk of stroke/cerebrovascular accident (CVA) and dysphagia was similar in both groups (Fig. 3). Patients undergoing outpatient ACDF had reduced risk of wound complications (infection and/or dehiscence) (RR 0.59, 95% CI [0.52–0.68]; $n = 8$, $I^2 = 0.0\%$) and reduced need for blood transfusion (RR 0.21, 95% CI [0.09–0.48]; $n = 5$, $I^2 = 55.0\%$) (Fig. 4). The estimated blood loss (in ml) (WMD $-13.6$, 95% CI [$-20.2$ to $-7.0$]; $n = 4$, $I^2 = 72.1\%$) and the length of hospital stay (in hours) (WMD $-22.3$, 95% CI [$-40.1$ to $-4.5$]; $n = 6$, $I^2 = 100.0\%$) was also lesser in the outpatient group (Fig. S1). However, the risk of developing hematoma was similar between the groups (Fig. 4).

The risk of unplanned reoperation (RR 0.33, 95% CI [0.24–0.46]; $n = 9$, $I^2 = 19.0\%$), readmission (RR 0.57, 95% CI [0.46–0.70]; $n = 10$, $I^2 = 62.4\%$) and reintubation (RR 0.47, 95% CI [0.27–0.81]; $n = 5$, $I^2 = 0.0\%$) was substantially reduced in outpatient ACDF group (Fig. 5). While the risk of progressive renal insufficiency, acute renal failure, and cardiac complications was similar in both groups, the outpatient ACDF was associated with a reduced risk of pulmonary complications (pneumonia, pulmonary embolism, ventilator dependence) (RR 0.43, 95% CI [0.27–0.68]; $n = 9$, $I^2 = 88.7\%$) (Fig. 6). The risk of developing sepsis (RR 0.56, 95% CI [0.49–0.64]; $n = 6$, $I^2 = 0.0\%$) and urinary tract infection (UTI) (RR 0.34, 95% CI [0.20–0.60]; $n = 7$, $I^2 = 59.7\%$) was lower in the outpatient group, compared to inpatient subjects (Fig. 7).

## Publication bias and GRADE certainty of evidence

No evidence of publication bias was present for all outcomes except for the risk of stroke/CVA (egger $p$-value 0.01) and UTI (egger $p$-value 0.02). The funnel plots for the outcomes are presented in Figs. S2–S19, and the GRADE certainty of evidence is summarized in Table 2. For all outcomes, the certainty varied from "low" to "very low" primarily due to the inherent risk of bias, high heterogeneity, and concerns of indirectness.

**Table 2  Certainty of pooled evidence using the GRADE approach.**

| | Number of studies with design | Certainty of the evidence (GRADE) | Effect size (95% CI); $I^2$ | Reason for downgrading |
|---|---|---|---|---|
| Risk of "any" complications | $N = 14$ (All cohort) | Very low | RR 0.45 (0.35 to 0.57); 74.7% | Risk of bias; high inconsistency; concerns of indirectness |
| Risk of mortality | $N = 7$ (All cohort) | Low | RR 0.35 (0.16 to 0.77); 0.0% | Risk of bias; concerns of indirectness |
| Risk of deep vein thrombosis | $N = 8$ (All cohort) | Low | RR 0.56 (0.37 to 0.85); 39.6% | Risk of bias; concerns of indirectness |
| Risk of stroke/CVA | $N = 6$ (All cohort) | Very low | RR 0.88 (0.72 to 1.08); 0.0% | Risk of bias; concerns of indirectness; presence of publication bias |
| Risk of Dysphagia | $N = 8$ (All cohort) | Very low | RR 0.82 (0.56 to 1.21); 19.2% | Risk of bias; concerns of indirectness; imprecision |
| Risk of wound infection/dehiscence | $N = 8$ (All cohort) | Low | RR 0.59 (0.52 to 0.68); 0.0% | Risk of bias; concerns of indirectness |
| Risk of need for blood transfusion | $N = 5$ (All cohort) | Very low | RR 0.21 (0.09 to 0.48); 55.0% | Risk of bias; high inconsistency; concerns of indirectness |
| Risk of haematoma | $N = 5$ (All cohort) | Very low | RR 0.22 (0.04 to 1.16); 57.3% | Risk of bias; concerns of indirectness; imprecision; high inconsistency |
| Risk of unplanned reoperation | $N = 9$ (All cohort) | Low | RR 0.33 (0.24 to 0.46); 19.0% | Risk of bias; concerns of indirectness |
| Risk of readmission | $N = 10$ (All cohort) | Very low | RR 0.57 (0.46 to 0.70); 62.4% | Risk of bias; high inconsistency; concerns of indirectness |
| Risk of reintubation | $N = 5$ (All cohort) | Low | RR 0.47 (0.27 to 0.81); 0.0% | Risk of bias; concerns of indirectness |
| Risk of renal complications | $N = 5$ (All cohort) | Very low | RR 0.93 (0.47 to 1.85); 85.6% | Risk of bias; concerns of indirectness; high inconsistency; imprecision |
| Risk of pulmonary complications | $N = 9$ (All cohort) | Very low | RR 0.43 (0.27 to 0.68); 88.7% | Risk of bias; concerns of indirectness; high inconsistency |
| Risk of cardiac complications | $N = 7$ (All cohort) | Very low | RR 0.88 (0.58 to 1.32); 31.0% | Risk of bias; concerns of indirectness; imprecision |
| Risk of sepsis | $N = 6$ (All cohort) | Low | RR 0.56 (0.49 to 0.64); 0.0% | Risk of bias; concerns of indirectness |
| Risk of urinary tract infections (UTI) | $N = 7$ (All cohort) | Very low | RR 0.34 (0.20 to 0.60); 59.7% | Risk of bias; concerns of indirectness; presence of publication bias; high inconsistency |
| Estimated blood loss (ml) | $N = 4$ (All cohort) | Very low | MD −13.6 (−20.2 to −7.0); 72.1% | Risk of bias; concerns of indirectness; high inconsistency |
| Duration of hospital stay (hours) | $N = 6$ (All cohort) | Very low | MD −22.3 (−40.1 to −4.5); 100.0% | Risk of bias; concerns of indirectness; high inconsistency |

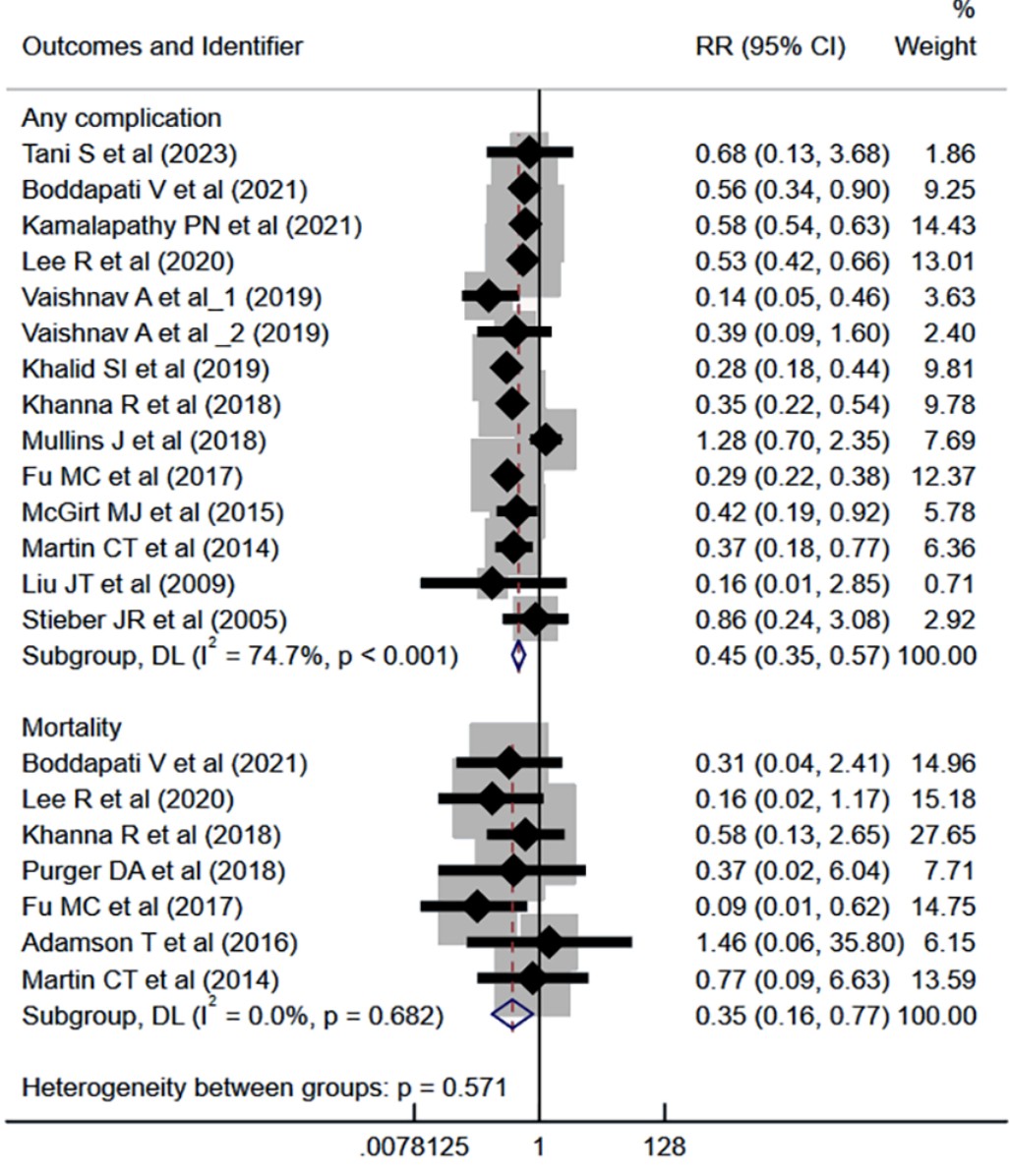

**Figure 2** Risk of "any" complications and mortality in those undergoing outpatient ACDF, compared to inpatient ACDF. Studies: *Tani et al., 2023; Boddapati et al., 2021; Kamalapathy et al., 2021; Lee et al., 2021; Vaishnav et al., 2019a; Vaishnav et al., 2019b; Khalid et al., 2019; Khanna et al., 2018; Mullins et al., 2018; Fu et al., 2017; McGirt et al., 2015; Martin et al., 2014; Liu, Briner & Friedman, 2009; Stieber et al., 2005; Purger et al., 2019; Adamson et al., 2016.*

## DISCUSSION

This meta-analysis synthesizes evidence from 21 studies encompassing 164,541 patients, comparing the outcomes of outpatient and inpatient ACDF. The findings demonstrate that outpatient ACDF correlates with a significantly lower risk of overall complications,

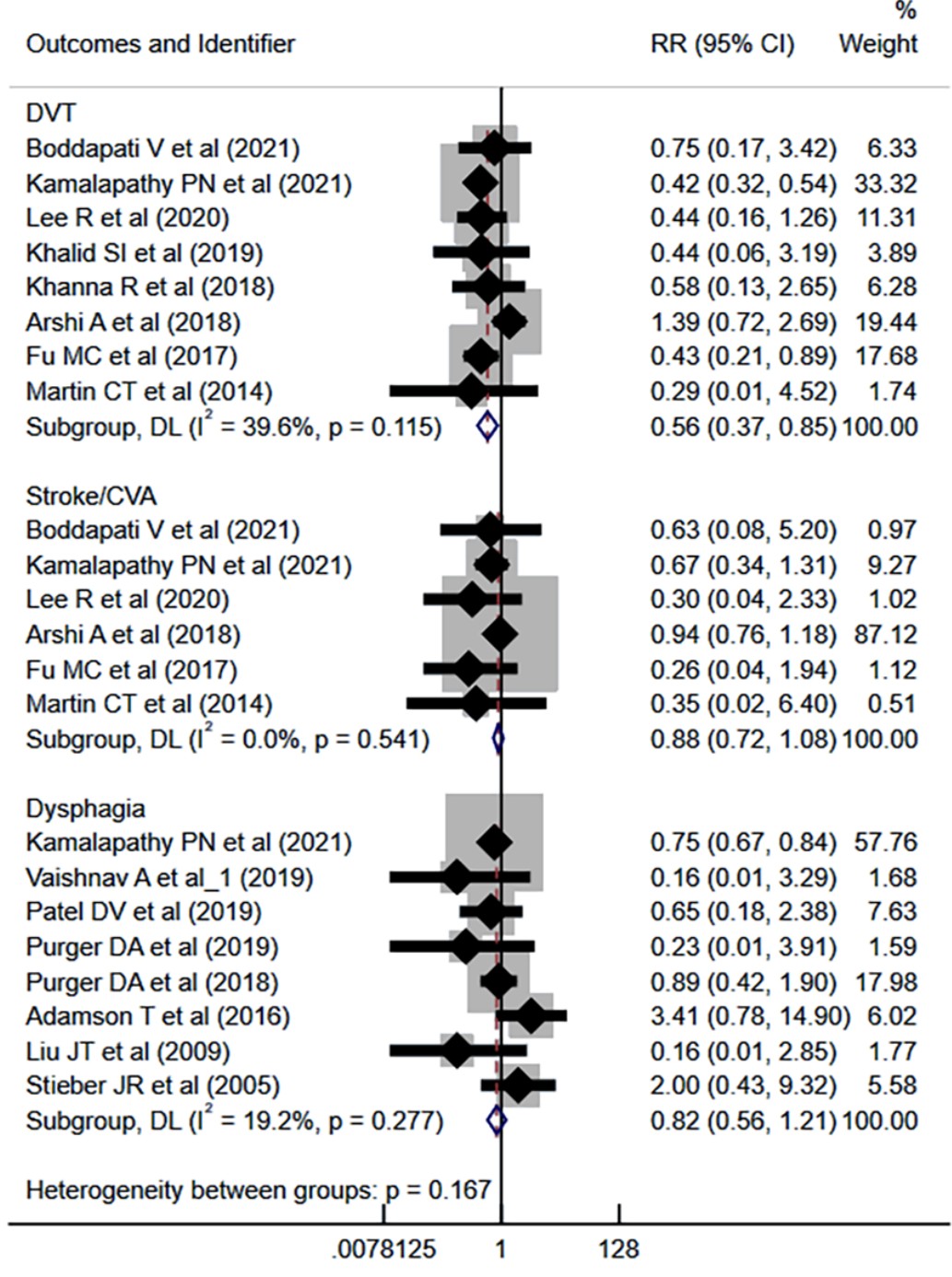

**Figure 3** **Risk of deep vein thrombosis (DVT), stroke or cerebrovascular accidents (CVA) and dysphagia in those undergoing outpatient ACDF, compared to inpatient ACDF.** Studies: *Boddapati et al., 2021*; *Kamalapathy et al., 2021*; *Lee et al., 2021*; *Vaishnav et al., 2019a*; *Khalid et al., 2019*; *Khanna et al., 2018*; *Arshi et al., 2018*; *Mullins et al., 2018*; *Fu et al., 2017*; *McGirt et al., 2015*; *Martin et al., 2014*; *Liu, Briner & Friedman, 2009*; *Stieber et al., 2005*; *Purger et al., 2018*; *Purger et al., 2019*; *Adamson et al., 2016*; *Lee et al., 2021*; *Patel et al., 2019*.

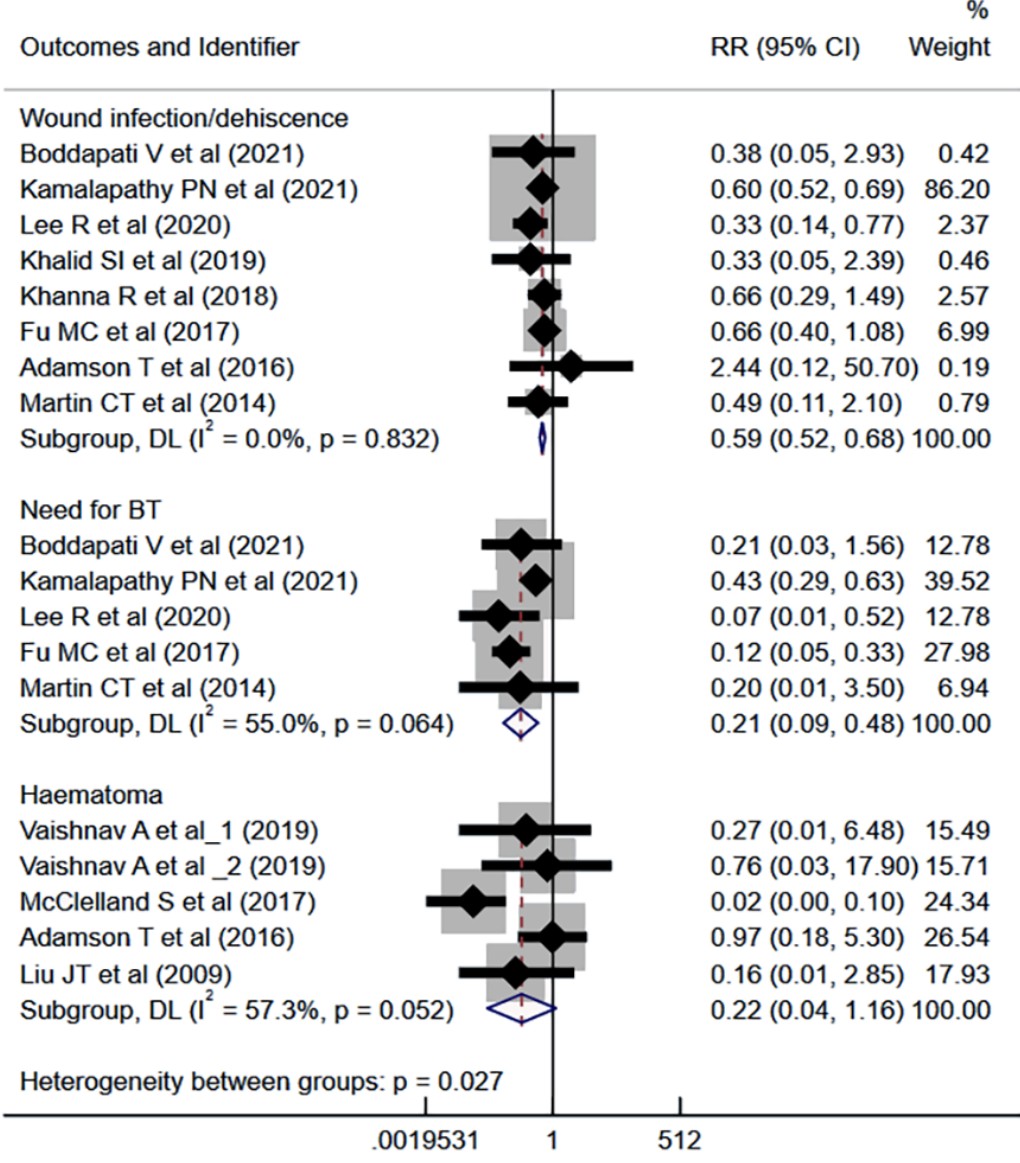

**Figure 4** Risk of wound complications, need for blood transfusion and haematoma in those undergoing outpatient ACDF, compared to inpatient ACDF. Studies: *Boddapati et al., 2021*; *Kamalapathy et al., 2021*; *Lee et al., 2021*; *Vaishnav et al., 2019a*; *Vaishnav et al., 2019b*; *Khalid et al., 2019*; *Khanna et al., 2018*; *Mullins et al., 2018*; *Fu et al., 2017*; *Martin et al., 2014*; *Liu, Briner & Friedman, 2009*; *Adamson et al., 2016*; *McClelland et al., 2017*.

mortality, DVT, wound complications, and the need for blood transfusion. Additionally, outpatient ACDF was linked to a lower length of hospital stay, estimated blood loss, and lower risks of unplanned reoperation, readmission, reintubation, incidences of pulmonary complications, sepsis, and UTI. However, risks of stroke/CVA, dysphagia, hematoma, and renal and cardiac complications were comparable between the two groups.

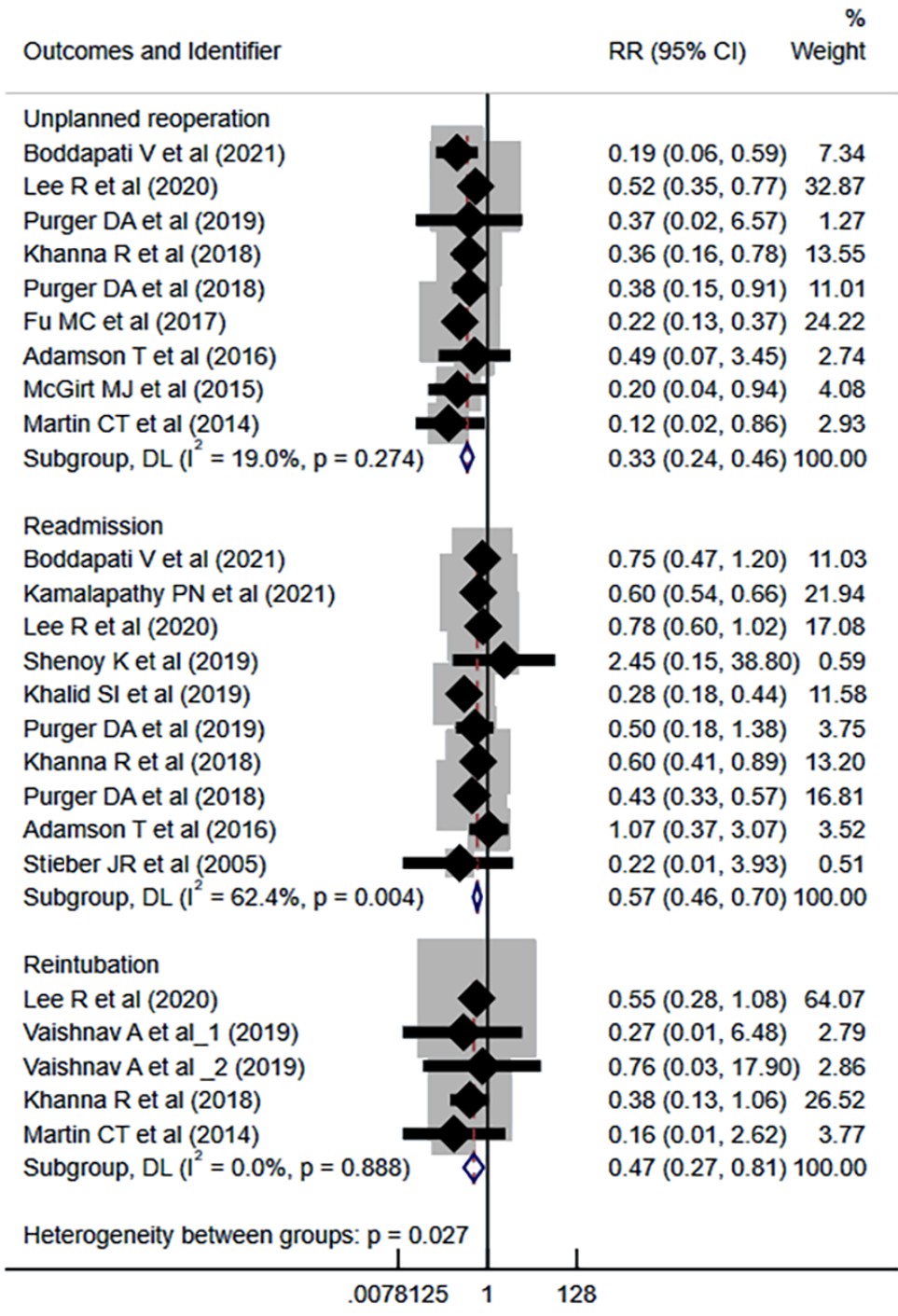

**Figure 5** Risk of unplanned reoperation, readmission and reintubation in those undergoing outpatient ACDF, compared to inpatient ACDF. Studies: *Boddapati et al., 2021*; *Kamalapathy et al., 2021*; *Lee et al., 2021*; *Vaishnav et al., 2019a*; *Vaishnav et al., 2019b*; *Khalid et al., 2019*; *Khanna et al., 2018*; *Fu et al., 2017*; *McGirt et al., 2015*; *Martin et al., 2014*; *Stieber et al., 2005*; *Purger et al., 2018*; *Adamson et al., 2016*; *Shenoy et al., 2019*; *Purger et al., 2019*.

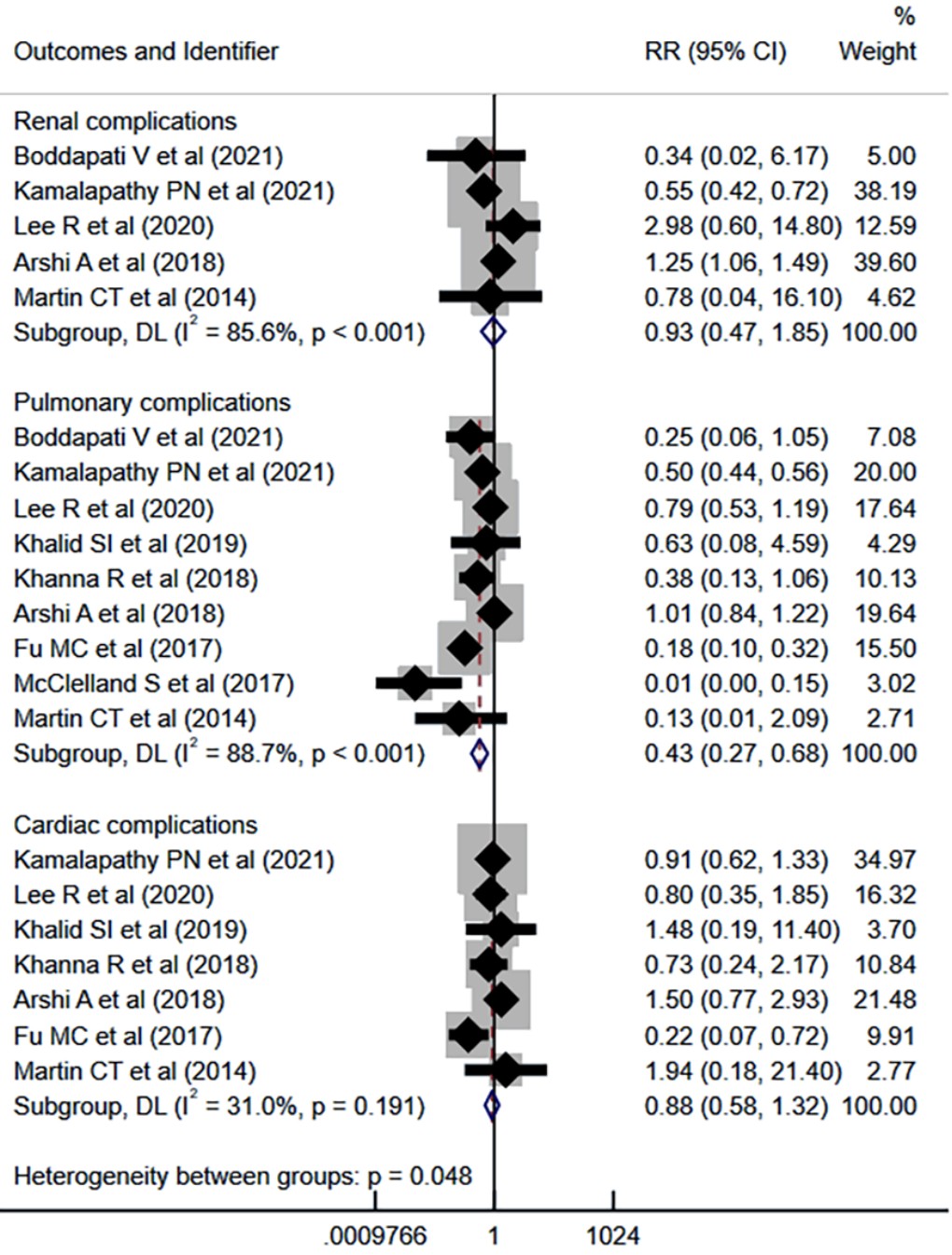

Figure 6 **Risk of renal, pulmonary and cardiac complications in those undergoing outpatient ACDF, compared to inpatient ACDF.** Studies: *Boddapati et al., 2021*; *Kamalapathy et al., 2021*; *Lee et al., 2021*; *McClelland et al., 2017*; *Khalid et al., 2019*; *Khanna et al., 2018*; *Arshi et al., 2018*; *Fu et al., 2017*; *McGirt et al., 2015*; *Martin et al., 2014*; *Stieber et al., 2005*; *Purger et al., 2018*; *Adamson et al., 2016*; *Shenoy et al., 2019*; *Purger et al., 2019*.

Our findings align with earlier reviews, including those by *Ban et al. (2016)* and *Yerneni et al. (2020)*, which highlighted the safety of outpatient ACDF for appropriately selected

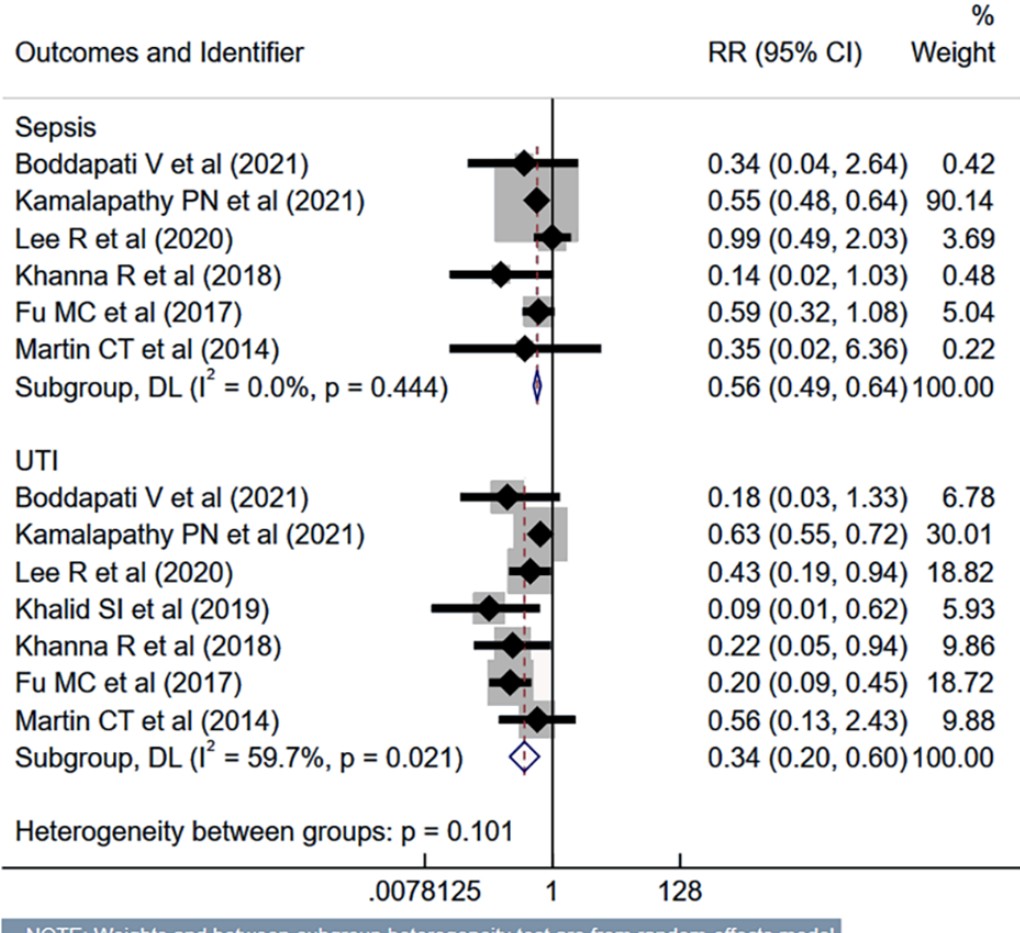

**Figure 7** **Risk of sepsis and urinary tract infection in those undergoing outpatient ACDF, compared to inpatient ACDF.** Studies: *Boddapati et al., 2021*; *Lee et al., 2021*; *Kamalapathy et al., 2021*; *Khalid et al., 2019*; *Khanna et al., 2018*; *Fu et al., 2017*; *Martin et al., 2014*.

patients. However, this study updates the field by incorporating ten additional recent studies and examining a broader spectrum of outcomes, thereby providing a more comprehensive evaluation of outpatient ACDF safety.

While this meta-analysis demonstrated the safety of ACDF as an outpatient procedure, it is important to take into account the quality of the evidence while interpreting the results. The mean NOS score of the studies was 7.2, indicating moderate quality. All studies were retrospective in nature and hence, the possibility of selection bias, *i.e.,* healthier patients being chosen for outpatient procedures cannot be neglected. This was evident in the comparison of baseline characteristics wherein the outpatient cohort was younger and had fewer comorbidities. In some studies (*Adamson et al., 2016*), outpatient procedure was conducted, initially, only in those patients who had the lowest risk of postoperative complications and gradually extended to more complex cases. *Boddapati et al. (2021)* have reported that in the USA, in 2018, only 32.9% of three-level ACDF were conducted in

an outpatient setting and surgeons still carried out four-level procedures in an in-patient setting, given the more complex nature of the surgery and higher tendency of complications. There has been an increasing trend of conducting two-level and three-level procedures in an outpatient setting, however, data is still limited (*Boddapati et al., 2021*). The study by *Arshi et al. (2018)* also reported that high-risk cases like elderly patients or those undergoing multi-level procedures still have significantly higher complication rates in an outpatient setting. This suggests that patient selection is still critical while deciding to perform ACDF as an outpatient or an inpatient procedure. Baseline comorbidities, age, ASA grade, *etc* would remain essential factors in selection of patients till more quality evidence from randomized controlled trials is made available. The present study could not conduct a subgroup analysis for high-risk patients due to lack of data. We believe that future studies should stratify patients as high-risk and low-risk categories to better assess the safety of ACDF as an outpatient procedure.

Many factors may explain the observed associations between outpatient ACDF and reduced complication rates. Patient selection plays a critical role, as outpatient ACDF patients are typically younger, healthier, and have fewer comorbidities, which collectively lower their baseline risk of complications. Additionally, outpatient facilities often adhere to enhanced perioperative protocols to ensure rapid recovery and early discharge, thereby reducing risks such as infection and sepsis (*Smith et al., 2020*). The shorter hospital stays associated with outpatient procedures could further minimize exposure to hospital-acquired infections, including pneumonia and UTIs. Lastly, the surgical and anesthetic stress experienced by outpatient ACDF patients is generally lower due to shorter procedures, reduced blood loss, and decreased anesthetic exposure, which may help reduce risks of DVT and pulmonary embolism (*Phan et al., 2017*).

This study has some limitations. There was substantial ($I^2 > 70\%$) heterogeneity in some outcomes, such as length of stay and estimated blood loss, likely reflecting differences in study populations, definitions of outcomes, and reporting standards. Additionally, the retrospective cohort design of all included studies introduces inherent risks of bias, such as unmeasured confounders and potential selection bias. For example, healthier patients are more likely to undergo outpatient procedures, skewing results in favor of outpatient ACDF.

We could not conduct a more detailed subgroup analysis in our study stratifying patients based on age, comorbidities, and complexity of surgery as segregated data was unavailable. Furthermore, the GRADE assessment revealed that the certainty of the evidence was low to very low for most outcomes due to high heterogeneity, risk of bias, and indirectness of the evidence. These limitations underscore the need for high-quality prospective studies or randomized clinical trials to strengthen the evidence base.

While our findings support the safety of outpatient ACDF, clinicians should consider patient-specific factors such as age, comorbidities, and surgical complexity when determining the appropriate surgical setting. The reduced risk of complications and shorter recovery times in outpatient settings suggest that this approach might be a cost-effective and patient-centered option if appropriate perioperative care protocols are followed. Given the low certainty of evidence on GRADE, high-quality prospective

cohort studies and randomized controlled trials are needed to provide reliable comparisons between outpatient and inpatient settings. Standardizing outcome definitions and reporting protocols will ensure consistency across studies. Studies should stratify outcomes based on the number of comorbidities and complexity of the case so that the benefits of out-patient surgery are clearly delineated. Additionally, future investigations should explore long-term outcomes, such as fusion success rates and chronic complications, to comprehensively understand the risks and benefits of outpatient ACDF. Research into the biological and clinical mechanisms underlying outcome differences could offer valuable insights to optimize surgical practices.

## CONCLUSION

This meta-analysis suggests that outpatient ACDF may be associated with fewer complications as compared to inpatient procedures in carefully selected patients. However, the findings should be interpreted cautiously as they are based on retrospective cohort studies, which are inherently prone to selection bias, and have a low to very low certainty of evidence. While outpatient ACDF demonstrates promising outcomes, careful patient selection and adherence to standardized perioperative protocols remain critical to ensure safety and efficacy. In the absence of robust randomized controlled trials, evidence from retrospective studies remains speculative.

### Funding
The authors received no funding for this work.

### Competing Interests
The authors declare there are no competing interests.

### Author Contributions

- Lili Ding conceived and designed the experiments, performed the experiments, analyzed the data, prepared figures and/or tables, authored or reviewed drafts of the article, and approved the final draft.
- Mengzhu Yin conceived and designed the experiments, performed the experiments, analyzed the data, authored or reviewed drafts of the article, and approved the final draft.
- Wenhua Yuan performed the experiments, analyzed the data, prepared figures and/or tables, authored or reviewed drafts of the article, and approved the final draft.

### Data Availability

The raw data is available in the Supplementary File.

## Supplemental Information

Supplemental information for this article can be found online at http://dx.doi.org/10.7717/peerj.20045#supplemental-information.

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
