# Peer review of "Safety of outpatient vs. inpatient anterior cervical discectomy and fusion: a systematic review and meta-analysis"

_PeerJ, doi:10.7717/peerj.20045_

## Round 0.1 · original submission · Major Revisions

· Academic Editor

Major Revisions

Reviewer 1 ·

Basic reporting

- Language and Structure: The manuscript is written in clear, professional English and is well-structured, following the standard format for systematic reviews and meta-analyses. Figures and tables are appropriately labeled and relevant.
- Literature and Context: The background is comprehensive and includes relevant references. However, some critical literature expressing concerns about outpatient ACDF in higher-risk patients (e.g., elderly, multilevel procedures) is underemphasized. For instance, Arshi et al. (2018), which reported higher revision and complication rates in outpatient settings, is not adequately discussed in the Discussion section, despite being cited.
- Raw Data: The manuscript includes supplementary figures and GRADE assessments; however, a full raw dataset extraction table (study-level outcomes) would aid reproducibility.

Suggested improvements:
- Better contextualization of conflicting findings (e.g., Arshi et al.).
- Include a supplementary appendix with outcome-level extracted data per study for transparency.

Experimental design

- Research Question: The research question is appropriate and well-articulated. The study aims to fill an evidence gap by updating prior meta-analyses with recent studies.
- Methodology: The search strategy, inclusion/exclusion criteria, and statistical methods are mostly appropriate. The use of PRISMA, GRADE, and Newcastle-Ottawa Scale strengthens the methodological rigor.
- Limitations: The authors acknowledge key limitations (retrospective design, heterogeneity, selection bias), but do not sufficiently explore the consequences of these issues. The Methods lack detail on how differences in surgical complexity or comorbidities were adjusted statistically (e.g., meta-regression or subgroup analysis).

Suggested improvements:
- Clarify whether any sensitivity or subgroup analyses were performed to assess impact of comorbidities, age, or surgical complexity.
- Discuss the potential confounding effect of healthier patients being preferentially selected for outpatient surgery.

Validity of the findings

- Data Robustness: The pooled estimates suggest outpatient ACDF is associated with fewer complications. However, the authors admit the GRADE certainty is low to very low, driven by high heterogeneity (I² often > 70%) and retrospective nature.
- Interpretation Concerns:
- The conclusion that “Outpatient ACDF significantly reduces complications” is overstated and not warranted based on the GRADE assessment.
- Selection bias likely drives much of the observed difference, as outpatient candidates were younger, healthier, and had fewer comorbidities in nearly all included studies.
- No randomized or prospective comparative data were included.

Suggested improvements:
- The conclusions should be moderated. For example, instead of stating outpatient ACDF “significantly reduces complications,” it would be more accurate to say, “is associated with fewer reported complications, although this may reflect selection bias.”
- Acknowledge that without randomized or well-controlled prospective studies, causal inferences are speculative.

Additional comments

This review is timely and well-organized, but conclusions should be tempered given the methodological constraints. Importantly, certain high-risk populations (elderly, high ASA scores, multi-level procedures) may still be better suited for inpatient care. The authors should consider:
- Discussing risks of under-triaging higher-risk patients to outpatient settings.
- Suggesting a stratified protocol for outpatient selection criteria in future research.

Reviewer 2 ·

Basic reporting

The abstract is structured; the keywords should be checked in accordance with MeSH.
The introduction should focus more on outpatient/inpatient specific frameworks and remote care principles in relation to the scientific literature (for e.g.Moldovan, F.; Moldovan, L. An Innovative Assessment Framework for Remote Care in Orthopedics. Healthcare 2025, 13, 736. doi: 10.3390/healthcare13070736). The objectives should be structured as research questions (RQ).

Experimental design

There is a lack of detail on how patients were deemed "suitable" for outpatient ACDF; please summarize selection protocols from included studies (e.g., ASA I-II, no severe myelopathy) to inform clinical applicability. Also, explicitly call for RCTs to validate findings, given the low GRADE certainty. Please indicate in the methodology who performed the search with initials.

Validity of the findings

The study’s conclusions are based on retrospective data with significant heterogeneity and potential biases; high heterogeneity (e.g., I² >70% for some outcomes) suggests variability in patient demographics, surgical techniques, and outcome definitions. Future research directions should be better emphasized at the end of the discussion section.

Additional comments

Please provide the Figures 2-6 in table format.
The references should be extended as suggested above.

---

## Round 0.2 · accepted · Accept

· Academic Editor

Accept

Thank you for your efforts in addressing the reviewer feedback. You have made satisfactory changes to the manuscript and it is now accepted for publication.

Reviewer 2 ·

Basic reporting

-

Experimental design

-

Validity of the findings

-

Additional comments

The authors have improved their paper accordingly.